# Network-Based Genetic Profiling Reveals Cellular Pathway Differences Between Follicular Thyroid Carcinoma and Follicular Thyroid Adenoma

**DOI:** 10.3390/ijerph17041373

**Published:** 2020-02-20

**Authors:** Md. Ali Hossain, Tania Akter Asa, Md. Mijanur Rahman, Shahadat Uddin, Ahmed A. Moustafa, Julian M. W. Quinn, Mohammad Ali Moni

**Affiliations:** 1Department of Computer Science & Engineering, Manarat International University, Khagan, Dhaka 1343, Bangladesh; ali.cse.bd@gmail.com; 2Electrical and Electronic Engineering, Islamic University, Kushtia 7005, Bangladesh; tania.eee.iu@gmail.com; 3Computer Science & Engineering, Jatiya Kabi Kazi Nazrul Islam University, Mymensingh 2205, Bangladesh; mijanjkkniu@gmail.com; 4Complex Systems Research Group & Project Management Program, Faculty of Engineering, The University of Sydney, Sydney, NSW 2006, Australia; shahadat.uddin@sydney.edu.au; 5Marcs Institute for Brain and Behaviour and School of Psychology, Western Sydney University, Sydney, NSW 2751, Australia; a.moustafa@westernsydney.edu.au; 6Bone Biology Divisions, Garvan Institute of Medical Research, Darlinghurst, NSW 2010, Australia; j.quinn@garvan.org.au; 7WHO Collaborating Centre on eHealth, School of Public Health and Community Medicine, Faculty of Medicine, The University of New South Wales, Sydney, NSW 2052, Australia

**Keywords:** thyroid cancer, carcinoma, protein-protein interaction, reporter transcription factors, reporter microRNAs, molecular pathways

## Abstract

Molecular mechanisms underlying the pathogenesis and progression of malignant thyroid cancers, such as follicular thyroid carcinomas (FTCs), and how these differ from benign thyroid lesions, are poorly understood. In this study, we employed network-based integrative analyses of FTC and benign follicular thyroid adenoma (FTA) lesion transcriptomes to identify key genes and pathways that differ between them. We first analysed a microarray gene expression dataset (Gene Expression Omnibus GSE82208, n = 52) obtained from FTC and FTA tissues to identify differentially expressed genes (DEGs). Pathway analyses of these DEGs were then performed using Gene Ontology (GO) and Kyoto Encyclopedia of Genes and Genomes (KEGG) resources to identify potentially important pathways, and protein-protein interactions (PPIs) were examined to identify pathway hub genes. Our data analysis identified 598 DEGs, 133 genes with higher and 465 genes with lower expression in FTCs. We identified four significant pathways (one carbon pool by folate, p53 signalling, progesterone-mediated oocyte maturation signalling, and cell cycle pathways) connected to DEGs with high FTC expression; eight pathways were connected to DEGs with lower relative FTC expression. Ten GO groups were significantly connected with FTC-high expression DEGs and 80 with low-FTC expression DEGs. PPI analysis then identified 12 potential hub genes based on degree and betweenness centrality; namely, TOP2A, JUN, EGFR, CDK1, FOS, CDKN3, EZH2, TYMS, PBK, CDH1, UBE2C, and CCNB2. Moreover, transcription factors (TFs) were identified that may underlie gene expression differences observed between FTC and FTA, including FOXC1, GATA2, YY1, FOXL1, E2F1, NFIC, SRF, TFAP2A, HINFP, and CREB1. We also identified microRNA (miRNAs) that may also affect transcript levels of DEGs; these included hsa-mir-335-5p, -26b-5p, -124-3p, -16-5p, -192-5p, -1-3p, -17-5p, -92a-3p, -215-5p, and -20a-5p. Thus, our study identified DEGs, molecular pathways, TFs, and miRNAs that reflect molecular mechanisms that differ between FTC and benign FTA. Given the general similarities of these lesions and common tissue origin, some of these differences may reflect malignant progression potential, and include useful candidate biomarkers for FTC and identifying factors important for FTC pathogenesis.

## 1. Introduction

Thyroid cancers are the most common type of endocrine malignancy, although they have a relatively low mortality rate compared to most other common metastatic diseases. The United States had 56,460 new diagnoses of thyroid cancer and 1780 related deaths reported in 2012 [1]. The incidence of thyroid cancers is also rising globally at about 5% per year, although some of this increase may be due to improved detection, and it notably affects those in the 20 to 34 year age range [2]. Thyroid cancers include several major types, such as papillary thyroid carcinomas, medullary thyroid carcinoma, anaplastic thyroid carcinoma, and follicular thyroid carcinomas (FTCs) [3]; FTC is one of the more aggressive types, although it accounts for a minority (14%) of total thyroid cancers [4,5,6].

The causes and cellular processes underlying FTC and controlling these tumours’ behaviours are poorly understood; accordingly, these cancers have few effective treatment options [7]. There is therefore a great need to understand the mechanisms that drive development and progression in FTC to identify new approaches to detection, estimate the risk of progression, and find new therapies. In addition, differential diagnosis of FTC is problematic as it can be difficult to distinguish from follicular thyroid adenoma (FTA), a benign and non-invasive lesion. Accordingly, there is more focus on molecular markers that distinguish FTC and FTA (and other types of thyroid lesions). Thus, Wojtas et al. conducted a gene expression comparison of FTC and FTA lesions which identified potential markers that can distinguish FTC from FTA with a sensitivity and specificity of 78% and 80%, respectively [8]. We aim to also use this dataset to identify pathways with different levels of activity in benign and aggressive thyroid tumours (here, FTA and FTC) that may reflect important molecular mechanisms that underlie their behaviour.

For such pathway studies, our starting point is the differential expression of genes (DEGs) between these lesions. We can then use analytic tools to study these DEGs and discover pathways, and pathway hub genes that may affect cell functions and thereby underlie tumour morbidity, growth, and invasion [9,10,11,12]. The repertoire of candidate pathway factors can be extended using approaches such as gene ontology analysis and protein-protein interaction (PPI) studies.

To identify such DEGs, microarray gene expression profiling is widely used [13,14], and several such studies have been performed on thyroid cancer subtypes [15]. For example, Huang et al. [16] studied gene expression profiles of thyroid cancers, although not functional interactions between gene products. To better understand the underlying molecular mechanisms and identify functionally critical biomolecules, integrative analysis within a gene network context is needed [17,18,19,20]. That may lead to candidate FTC biomarkers (the focus of the work by Wojtas et al.), but our main interest here is to obtain key or hub genes distinguishing malignant FTC from benign FTA, as these may be potential therapeutic targets. Such a systems biology approach integrating statistical network and topological analyses of experimental datasets can thus clarify disease mechanisms that are difficult to identify by other approaches [21]. Thus, we used a systems approach (Figure 1) to identify FTC molecular signatures at the miRNA and mRNA levels (and derived from these, the protein-protein interactions) that distinguish FTC tissue from FTA. These analyses should be more meaningful than a comparison between FTC with normal thyroid tissue, as the the two tumour types are similar, while markedly differing in their potential for invasive metastasis.

## 2. Materials and Methods

In this study, the multi-step analysis method we developed and applied is shown in Figure 1. We statistically analysed gene expression datasets to identify the DEGs and their regulatory patterns. We employed these DEGs to identify enriched pathways, biological processes, and annotation terms (i.e., Gene Ontology terms) by using functional enrichment methods. Then, to identify reporter biomolecules, we integrated the intermediate analysis results with biomolecular networks.

### 2.1. Dataset Employed and Statistical Methods Used

We obtained the gene expression data of FTC and FTA (GSE82208) for our study from the NCBI Gene Expression Omnibus (GEO) (http://www.ncbi.nlm.nih.gov/geo/) [8,22]. This dataset contains analyses of RNA from frozen tumour tissue specimens from 27 FTC and 25 FTA lesions using Affymetrix human genome U133 (Plus 2.0) arrays. The lesions were diagnosed histologically, many with a second diagnosis to confirm, when paraffin embedded material was available [8]. Gene expression analysis using microarrays is a widely used method to develop and refine the molecular determinants of disorders affecting humans. Using these methods, we analysed the gene expression profiles of FTC and FTA. To identify the DEGs between FTC samples and FTA, we ran *t*-tests using the Limma package [23]. To identify the up-regulated genes, we used the conditions of *p*-value <0.05 and logFC>2 (FC, fold change), and for identifying the down-regulated genes, *p*-value <0.05 and logFC<−2 were used. All identified up-regulated genes and down-regulation genes were considered DEGs. We applied the topological and neighbourhood-based benchmark methods to find gene-gene associations. A gene-gene network was constructed by using the gene-gene associations, where the nodes in the network represent gene [18,24]. This network can also be characterised as a bipartite graph. These topological and neighbourhood-based benchmark methods were adopted from our previous studies [12].

The common neighbours are based on the Jaccard Coefficient method, where the edge prediction score for the node pair is as [25]:(1)$$E\left(i,j\right)=\frac{N(G_{i}\cap G_{j})}{N(G_{i}\cup G_{j})}$$
where *G* is the set of nodes and *E* is the set of all edges. We used R software packages “comoR” [9] and “POGO” [12] to cross-check the genes-diseases associations.

#### 2.1.1. Functional Enrichment of Gene Sets

We performed gene ontology and pathway analysis on identified up-regulation genes and down-regulation genes using DAVID bioinformatics resources (https://david-d.ncifcrf.gov/) (version v6.8) [26] to obtain further insight into the molecular pathways that differ between FTC and FTA. In these analyses, GO and KEGG pathway databases were used as annotation sources. Enrichment results showing adjusted *p*-values <0.05 were considered significant.

#### 2.1.2. Construction and Analysis of Protein-Protein Interaction (PPI) Subnetworks

The PPI network was first constructed with the DEGs and analysed using STRING [27], a web-based visualisation software resource. The constructed PPI network was represented as an undirected graph, where nodes represent the proteins and the edges represent the interactions between the proteins. To construct the PPI network from the STRING database (http://string-db.org) [27], we used database data, data mined from PubMed abstract text. Co-expression, gene fusion, and neighbourhood were active interaction sources and a combined score that was greater than 0.4 was set as the level of significance. The PPI network was then visualised and analysed using Cytoscape (v3.5.1) [28,29]. A topological analysis was applied to identify highly connected proteins (i.e., hub proteins) by using the Cyto-Hubba plugin [30] where betweenness centrality and higher degree were employed. The top three modules (i.e., the three most highly interconnected protein clusters) in the PPI subnetwork were identified using the MCODE plug-in [30]. These modules were further analysed and characterised using enrichment analyses by NetworkAnalyst [31]. The KEGG pathway enrichment analysis of the PPI networks involving DEGs were performed by NetworkAnalyst [31].

#### 2.1.3. Identifying TFs and miRNAs that Influence the Expression of Candidate Genes

To identify TFs and miRNAs that affect transcript levels around which significant changes occur at the transcriptional level, we obtained experimentally verified TF-target genes from the JASPAR database [32] and miRNA-target gene interactions from TarBase [33] and miRTarBase [34] by using NetworkAnalyst tools [31] where betweenness centrality and higher degree filters were used. At the time, there were many techniques to measure the topological properties. We used the degree centrality (DC) and betweenness centrality (BC) to find out network’s topological properties. We can define the DC of a node *v* in a network as the total number of nodes which are directly connected to node *v* in that network. The definition can also be written as follows [35]:$$DC\left(v\right)=\sum_{j\in G}\frac{a_{vj}}{n-1}$$
whereas *n* represents total number of nodes in the network, and $a_{vj}$ represents that the node *v* and the node *j* are directly connected. In the case of betweenness centrality (BC), the total number of times of node *v* appearing in the shortest path between other nodes is quantified. It is also defined as follows [35]:$$BC\left(v\right)=\sum_{i\neq j\neq v\in V}\frac{\sigma_{ivj}}{\sigma_{ij}}$$
where $\sigma_{ij}$ = total number of shortest paths from node *i* to node *j*, and $\sigma_{ivj}$ = total number of paths through node *v*.

## 3. Results

### 3.1. Results of DEG Analyses

#### Transcriptomic Signatures: Differentially Expressed Genes

FTA and FTC tissue gene expression patterns were analysed using oligonucleotide microarrays from the NCBI GEO (http://www.ncbi.nlm.nih.gov/geo/query/acc.cgi?acc=GSE82208) [8]. Gene expression profiling was performed in 27 malignant FTCs and 25 FTAs: 598 genes were differentially expressed (p<0.05,>1.0 log2 fold change) relative to FTAs, of which 465 genes were significantly lower expression and 133 genes were higher expression levels in FTC lesions.

### 3.2. Pathway and Functional Correlation Analysis

By combining large scale and state of the art transcriptome and proteome analysis, we performed a regulatory analysis to gain further insight into the molecular pathways associated with the FTC and predicted links to pathways that differ relative to the benign FTAs. DEGs and pathways were analysed using KEGG pathway database (http://www.genome.jp/kegg/pathway.html) and functional annotation tool DAVID version 6.8 (http://niaid.abcc.ncifcrf.gov) to identify overrepresented pathway groups, and it was observed that four pathways were associated with DEGs with higher expression in FTC compared to FTA; namely, the “one carbon pool by folate” pathway, the p53 signalling pathway, the cell cycle, and the progesterone-mediated oocyte maturation signalling pathway. Fold enrichment, adjusted *p*-values, and genes associated with these pathways are presented in Table 1 (a).

It was also observed that eight pathways were significantly over-represented. Mineral absorption, thyroid hormone synthesis, pathways in cancer, oestrogen signalling, sphingolipid metabolism, protein processing in the endoplasmic reticulum, axon guidance, and focal adhesion pathways (Table 1 (a)) were identified. These are associated with the DEGs expressed at lower levels in FTC lesions compared to FTA. We then performed GO analysis using DAVID to obtain further insight into the molecular roles and biological function of DEGs identified in this study. From this analysis, nine GO groups were associated with DEGs that were more highly expressed in FTC (see Table 2 (a)). Reflecting their greater number, the genes with lower expression in FTC compared to FTA were associated with 80 GO groups; the 10 most significant GO groups are presented in Table 2 (b). Fold enrichment, adjusted *p*-values, and genes associated with these ontologies are also indicated.

#### 3.2.1. Proteomic Signatures: Hub Target Proteins from PPI Analysis

A PPI network was constructed using the DEGs identified in this study (Figure 2A) by using the STRING package. Topological analyses using STRING and further analysis by Cytoscape’s Cyto-Hubba plugin identified the most significant hub proteins, which were identified as gene products of TOP2A, JUN, EGFR, CDK1, FOS, CDKN3, EZH2, TYMS, PBK, CDH1, UBE2C, and CCNB2. The simplified PPI network for the most significant hub genes was constructed by using Cyto-Hubba plugin and is shown in Figure 2B.

#### 3.2.2. Enrichment Analysis of the Modules Found in PPI of DEG

The PPI network was analysed by using MCODE plug-in in Cytoscape (version 3.5.1), and top three modules were selected (Figure 3).

The enrichment analysis of the top three modules was analysed by using DAVID. Module 1 represented a set of biological process significantly enriched for ATP binding, cell division, and cell proliferation; the associated enriched cellular components were the kinetochore, the condensed chromosome kinetochore (both of which relate to cell division), and nucleoplasm and cytoplasm, which are very general categories. Molecular functions significantly enriched in Module 1 included chromatin binding and protein kinase activity (Table 3 (a)). Module 2 showed significantly enriched biological process positive regulation of protein phosphorylation and protein autophosphorylation; these significantly enriched cellular components included membrane raft, endosome, apical plasma membrane, and focal adhesion. Molecular functions of Module 2 were enriched in protein tyrosine kinase activity, enzyme binding, protein binding, and integrin binding (Table 3 (b)).

Module 3’s gene ontology annotation showed a significant enrichment for protein ubiquitination, with significantly enriched cellular components being ubiquitin ligase complex and SCF ubiquitin ligase complex, which have involvement in proteosomal and autophagy functions. Module 3’s molecular functions showed significantly enrichment in ubiquitin-protein transferase activity and zinc ion binding (Table 3 (b)). It is important to note that pathway enrichment analyses found Modules 1 and 2 were significantly enriched in protein binding pathways, which was not seen in Module 3 (Table 3 (b)).

### 3.3. Regulatory Signatures: TFs and miRNAs Affecting DEG Transcript Levels

#### 3.3.1. Transcriptional Regulatory Network Construction and TF Enrichment Analysis

The construction of reporter TFs and DEGs interaction network by NetworkAnalyst revealed a number of potentially important TFs selected by the topological analysis using a dual metric approach involving degree and betweenness (Figure 4). The top 10 ranked TFs with the highest degree and betweenness centrality included FOXC1, GATA2, YY1, FOXL1, E2F1, NFIC, SRF, TFAP2A, HINFP, and CREB1. Pathway enrichment analysis of the TFs that differ between FTC and FTA and DEGs mainly identified as statistically significant, a number of relevant pathways which included transcriptional misregulation in cancer, acute myeloid leukaemia, pathways in cancer, thyroid cancer, cell cycle, dorso-ventral axis formation, arrhythmogenic right ventricular cardiomyopathy (ARVC), cocaine addiction, and prostate cancer pathways (Table 4 (b)).

#### 3.3.2. miRNA Regulatory Network Construction and Enrichment Analysis of miRNA

We constructed TC DEG-miRNA interactions networks using NetworkAnalyst (Figure 5). By topological analysis, the top 10 ranked miRNAs by highest degree and betweenness centrality were selected, which included hsa-mir-335-5p, hsa-mir-26b-5p, hsa-mir-124-3p, hsa-mir-16-5p, hsa-mir-192-5p, hsa-mir-1-3p, hsa-mir-17-5p, hsa-mir-92a-3p, hsa-mir-215-5p, and hsa-mir-20a-5p. The analysis indicates that these are miRNAs having the highest potential to regulate levels of the DEG transcripts differently in FTC and FTA. The pathway enrichment analysis of the miRNA associated DEGs networks identified the p53 signalling pathway, dorso-ventral axis formation, pertussis, adherens junction, thyroid cancer, pathways in cancer, focal adhesion, sphingolipid metabolism, PPAR signalling pathway, and the ECM-receptor interaction signalling system as statistically significant (Table 4 (a))

## 4. Discussion

To clarify the molecular mechanisms underlying the pathological features (especially malignancy potential) that distinguish malignant and benign thyroid lesions, we examined differential gene expression between FTC and FTA. Such a comparison is likely to be more informative than making such a comparison of malignant and normal tissues, which would contain many more differences that relate to neoplastic cell development. In addition, the lesions contain significant non-neoplastic cell components (such as mesenchymally-derived and endothelial cells) that are also likely to differ markedly from those of normal tissue.

The DEGs identified from the FTC/FTA comparison were used to identify possible regulatory patterns, key molecular pathways, and protein-protein interactions among these pathway gene products. Such an analytical approach can be used to find molecular signatures that may serve either as potential therapeutic targets or as biomarkers to differentially diagnose or identify FTC. As did the original study by Wojtas et al. [8], we identified significantly altered genes that may be candidate biomarkers that distinguish FTC. However, another important use of such data (and our main focus here) is to identify and characterise biological functions associated with these genes that may give insights into the biology and behaviour of FTC lesions themselves. Wang et al. [36] examined on normal tissue vs. FTC and normal tissue vs. FTA in the microarray dataset accession number GSE27155. They identified some important pathways that included drug metabolism cytochrome p450, viral carcinogenesis, tyrosine metabolism, cytokine-cytokine receptor interaction, and cocaine Addition as significant pathways distinguishing normal from FTC tissue. Our analysis of the FTA vs. FTC tissue data found several important pathways, as described in the above results section. However, there were no pathways that we identified (in our FTC/FCA comparison) which were also seen in the studies comparing FTC to normal tissues. This almost certainly reflects the fact that a tumour has very different cellular profiles compared to normal tissue (e.g., infiltrating leukocytes and stromal cells) so in this circumstance, where we want to better understand the behaviour of the tumour, it is be better to compare with a benign or non-aggressive tumour such as FTA. Our study thus identified a number of important pathways that were differentially expressed in FTC, some of which would be expected by the malignant profile of FTC compared to FTA. For example, the p53 signalling pathway plays an important role in cancer cell apoptosis and DNA repair by causing cycle arrest in response to DNA damage [37], and has been previously implicated in thyroid cancers [38]. Cell cycle progression is regulated and facilitated by cyclin-dependent kinases that are activated by cyclins such as cyclin D1. Indeed, cyclin D1 levels can influence tumour progression, and may have prognostic significance in FTC [39]. Related to this, it has also been reported that the one-carbon metabolism pathway is actively involved in cancer progression, probably due to its involvement in nucleotide synthesis [40]. We have summarised the significant GO groups associated with the DEGs, as shown in Table 2.

PPI network reconstruction and the analysis of reconstructed PPI network represent a powerful approach for understanding disease mechanisms, so to construct a PPI network for the DEGs in our study, we combined results of statistical analyses with the protein interactome network. To identify potential hub proteins, topological analysis strategies were employed. This method identified 12 hub genes (Table 1 (b)), including a number of genes commonly associated with cancers (including thyroid cancers), such as EGRF, JUN, FOS, CDK1, and other cyclin pathway genes, E-cadherin (CDH1) and E2F1. The hub protein EGFR is linked to growth in many types of cancer, and cardiovascular diseases, and has been previously linked to thyroid cancers [41]. JUN and FOS are best characterised as subunits of the AP-1 transcription factor that has a crucial role in many types of cell differentiation and inflammation processes [42]. GO databases also delineated particular roles for JUN in angiogenesis and the regulation of endothelial cell proliferation, both of which have central roles in metastasis and invasion (Table 2 (b)). CDK1 is an important cell cycle regulator and is involved in breast, lung, and ovary carcinomas [43]. CDK1 overexpression has been documented in lung cancer, lymphoma, and advanced melanoma, while loss of cytoplasmic CDK1 predicts poor patient survival and may confer chemotherapeutic resistance in the latter [44]. Cyclin B1 (in the same family as CDK1) overexpression and/or mislocalisation has been described in several primary cancers, including thyroid carcinoma and colon, gastric, prostate, breast, and non small-cell lung cancers [44]. It is also actively involved in mitotic cell cycle phase transition (Table 2 (a)). CDKN3 encodes a cyclin inhibitor (regulating cell cycle) and has been described as being overexpressed in lung adenocarcinoma (ADC), squamous cell carcinoma (SCC), hepatocellular carcinoma, cervical cancer, and epithelial ovarian cancer [45]. Over-expression of EZH2 is frequently observed in many cancer types [46].

TYMS expression is also associated with the risk of development of epithelial cancers and lymphoma cancer [47]. In addition, in hereditary diffuse gastric cancer (HDGC), the hub gene CDH1 mutations are connected with an increased incidence of lobular carcinoma of the breast, and possibly, prostate cancer and colorectal carcinoma [48]. This gene is also involved in extracellular matrix organisation (Table 2 (b)). Hub gene CCNB2 is known to be underexpressed in thyroid cell tumours [49]. We found that CDK1 and CCNB2 hubs genes were actively associated with the p53 signalling, progesterone-mediated oocyte maturation, and cell cycle pathways. The hub gene TYMS is also involved in one carbon pool by folate pathway; FOS, JUN, and CDH1 genes are involved in pathways in cancer; and JUN and FOS genes are involved in the oestrogen signalling pathway. A number of other hub genes have been described as associated with malignant thyroid cancers, including EXH2, UBCH10,TFAP2A, TOP2A, and SRF [50,51,52,53,54]. By considering the possible roles of these hub proteins in pathogenesis of FTC and other related diseases, new roles for these proteins may be identified. In addition to the above-mentioned hub genes, we also identified a number of factors not previously noted as having a role in FTC or thyroid cancers, including UBA52, GATA2, FOXL1, and NFIC.

As the regulation of gene expression is controlled by TFs and miRNAs at post-transcriptional and/or transcriptional levels, changes in these molecules may provide crucial information regarding the dysregulation of gene expression in FTC. Thus, we investigated TFs and their relationship to DEGs in these tumours, including FOXC1. One prior study found that FOXC1 is strongly associated with thyroid cancers [55]. The overexpression of YY1 in differentiated thyroid cancers has also been noted [56]. The pathway analysis of these TFs showed statistically significance in FTC. From the DEGs, a miRNA interaction network of miRNAs (Table 1 (b)) was also identified and analysed. Our pathway analysis of these miRNA also showed statistically significant relationships with FTC. Notably, the target DEGs of these miRNAs and TFs included the p53 signalling pathway, PPAR signalling pathway, dorso-ventral axis formation pathway, focal adhesion pathway (Table 4 (a)), pathways in cancer, transcriptional misregulation in cancer, and thyroid cancer pathway. The miRNA species we identified that have previously been shown to have a role in thyroid cancer, including hsa-mir-335-5p, hsa-124-3p, hsa-mir-17-5p, and hsa-mir-20a-5p [57,58,59,60]. However, six other miRNAs we identified have not previously been associated with thyroid lesions. miRNA species have wide ranging effects on gene expression which are indirect, so further analysis for these are needed.

We analysed the dataset originally generated by Wojtas et al., who used combined micro-array methods and literature meta-analysis to achieve this to find markers to distinguish FTC and FTA. The gene markers they identified included TFF3, CPQ (previously characterised FTC markers), PLVAP, and ACVRL1, all genes expressed in the thyroid, although ACVR1 is an activin receptor with wide expression. These latter genes were not identified by our analytical approach, but it should be noted that our study used very different analytic tools, focussing on identifying pathways and on finding evidence for gene functions using PPI and other resources. Our analytical approach was somewhat similar to that of Wang et al. [36] who examined an older thyroid cancer dataset (GSE27155) containing 10 FTA, 14 FTC, and 4 normal thyroid tissue samples plus those of other types of thyroid lesions. While we employed a number of additional analytical tools (some developed previously by us), there were a number of significant genes and pathways that were also found in the work by Wang et al. [36] where they compared FTC and FTA. These notably included the AP-1 TF and its components (e.g., JUN and FOS), BCL2, factors relating to DNA damage, and cyclin-related factors that regulate the cell cycle. As noted above, We identified no genes that Wang et al. identified when comparing FTC and normal tissues. Recent studies by Shang et al. and by Liang and Sun [61] examined datasets generated from papillary thyroid cancer lesions that were compared with normal tissues. DEGs identified in these studies were analysed by methods related to our approach [35]. However, unlike the Wang et al. study cited above, very few of the hub genes these studies identified were found in our analysis, BCL2 being the main exception. This lack of concordance with the FTC studies may reflect the different tumour types (and lower malignancy rates) represented by papillary thyroid cancer, and it might be useful to compare all these datasets in further detail to determine how these types of tumour differ. We also note that work by Pfeifer et al. [62] developed a classification using gene expression data that considered only five genes, and this distinguished FTC from FTA with good accuracy. They identified five genes (ELMO1, EMCN, ITIH5, KCNAB1, and SLCO2A1) as biomarkers. However, it should be emphasised that the focus of our work and methodology is different (since we studied pathway differences not gene classifiers), so not surprisingly, there were no significant genes identified in common between our work and Pfeifer et al.

## 5. Conclusions

Our data-driven approach has uncovered a number of significant molecular mechanisms that may underlie the pathogenic differences seen between FTC and FTA. In this study we used integrated bioinformatic analyses to study gene expression profiles in FTC and FTA, and used this information to identify candidate hub proteins that could play significant roles in FTC development. Our results included a number candidate genes and pathways that indicate the directions for future experimental work needed to clarify the roles of these cellular factors. This study identified gene networks that advance our understanding of FTC pathogenesis and indicates new avenues to develop therapies for FTC.

## Figures and Tables

**Figure 1 ijerph-17-01373-f001:**
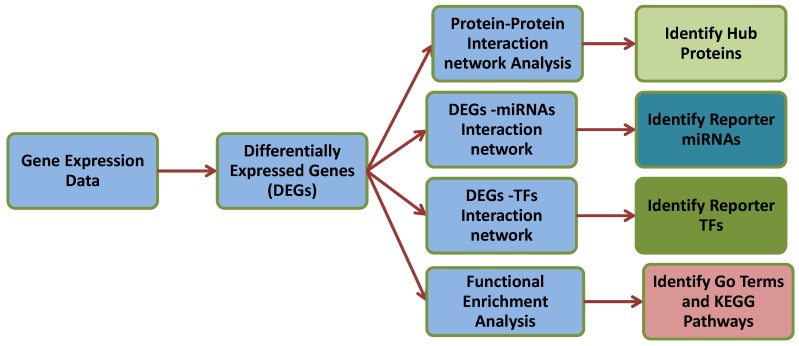
The multi-stage analysis methodologies employed in this study. Gene expression datasets related to follicular thyroid carcinoma (FTC) and follicular thyroid adenoma (FTA) tissues were collected from the NCBI Gene Expression Omnibus (NCBI-GEO) database and statistically analysed using GEO2R to identify differential expression of genes (DEGs). Four types of functional enrichment analyses of DEGs were then performed to identify significantly enriched pathways. Thus, we constructed protein-protein interaction networks around DEGs topological analyses to identify putative pathways hub proteins, identified possible micro-RNA (miRNA) and transcription factor (TF) interactors, and used Gene Ontology annotation terms to provide pathways enrichment. TF and miRNA studies employed JASPAR and miRTarbase databases, respectively. DEGs were integrated with those networks, and higher degree and the betweenness centrality were used to designate TFs and miRNAs as the reporter transcriptional regulatory elements. The target DEGs of reporter miRNAs and TFs were subjected to pathway enrichment analyses.

**Figure 2 ijerph-17-01373-f002:**
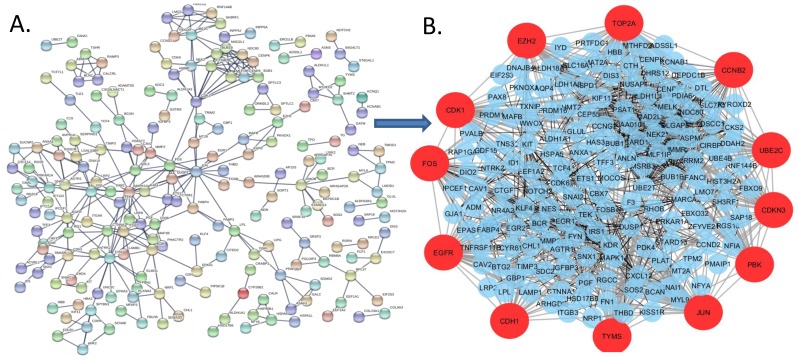
(**A**) The protein-protein interaction network constructed using the DEGs identified in the FTC/FTA comparison. (**B**) The protein-protein interaction(PPI) network identified in FTC/FTA dataset DEGs showing the most significant hub genes on the periphery (red).

**Figure 3 ijerph-17-01373-f003:**
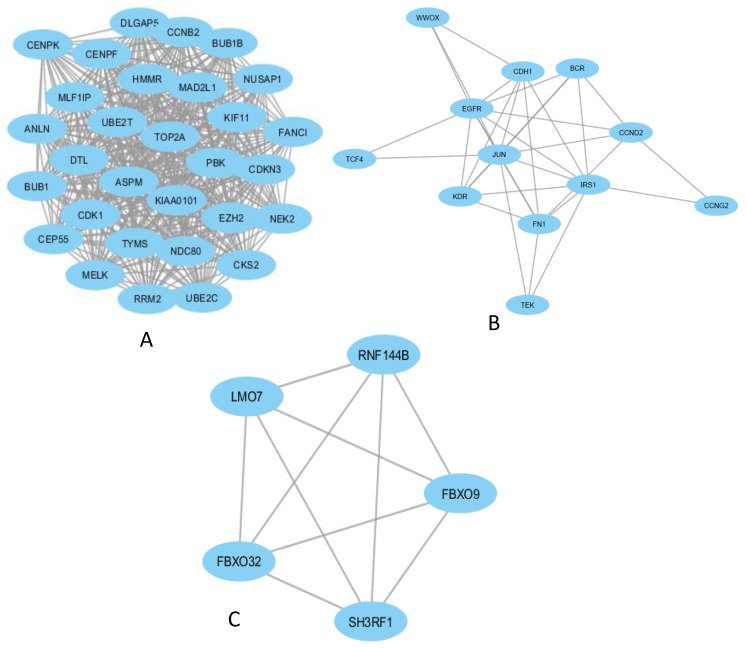
Top three modules in the protein-protein interaction network of the DEGs in TC. The nodes indicate the DEGs and the edges indicate the interactions between two genes. (**A**): Module-1; (**B**): Module-2; (**C**): Module-3.

**Figure 4 ijerph-17-01373-f004:**
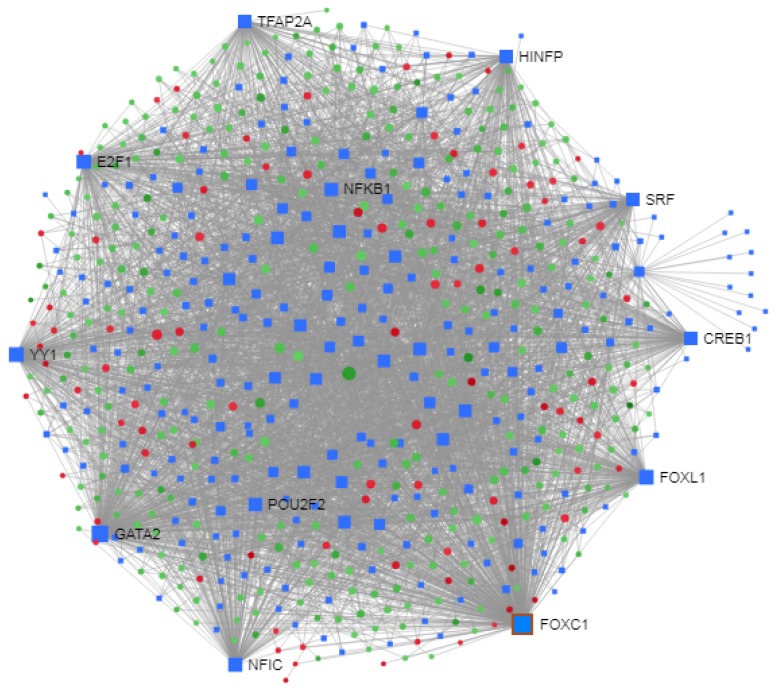
Construction of regulatory networks of TF-DEG interactions. Red nodes indicate up-regulation and green nodes represent down-regulated DEGs.

**Figure 5 ijerph-17-01373-f005:**
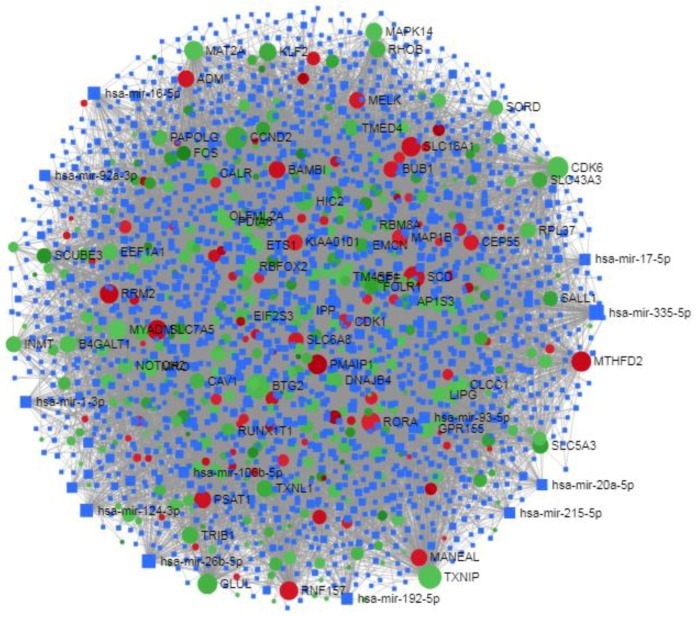
Construction of regulatory networks of the DEGs-miRNAs interaction. Red nodes indicate up-regulation and green nodes represent down-regulated DEGs.

**Table ijerph-17-01373-t001a:** (**a**)

KEGG ID	Pathway	Genes in the Pathway	Fold Enrichment	Adjusted *p*-Value
**A.**				
hsa00670	One carbon pool by folate	MTHFD2, TYMS, SHMT2, ALDH1L1	24	5.21E-04
hsa04115	p53 signaling pathway	CDK1, CCNB2, RRM2, PMAIP1, IGFBP3	9	2.03E-03
hsa04110	Cell cycle	CDK1, MAD2L1, CCNB2, BUB1, BUB1B	5	1.77E-02
hsa04914	Progesterone-mediated oocyte maturation	CDK1, MAD2L1, CCNB2, BUB1	6	3.31E-02
**B.**				
hsa04978	Mineral absorption	MT1M, MT2A, CYBRD1, MT1E, MT1H, MT1X, MT1G, MT1F	6	3.44E-04
hsa04918	Thyroid hormone synthesis	SLC26A4, TG, HSP90B1, PAX8, TPO, HSPA5, LRP2, TSHR, IYD	4	9.55E-04
hsa05200	Pathways in cancer	FGFR2, BCR, EPAS1, PGF, RXRB, GNAI1, RUNX1T1, FZD1, CDH1, CDK6, KIT, CTNNA1, MMP2, CXCL12, TCF7L1, AGTR1, FOS, HSP90B1, JUN, PAX8, SOS2, LAMB1	2	5.48E-03
hsa04915	Estrogen signaling pathway	HSPA1L, FOS, HSP90B1, GNAI1, JUN, SOS2, HSPA1A, HSPA1B, MMP2	3	8.41E-03
hsa00600	Sphingolipid metabolism	SPTLC1, SGMS2, SPTLC3, GALC, PLPP3, ASAH1	4	1.17E-02
hsa04360	Axon guidance	NRP1, LIMK2, SEMA6D, FYN, GNAI1, PLXNA2, SEMA3D, UNC5C, CXCL12, EPHA3	3	1.22E-02
hsa04141	Protein processing in endoplasmic reticulum	HSPA1L, HSP90B1, ERO1B, UBE4B, PDIA6, HSPA1A, HSPA1B, HSPA5, CALR, MAN1C1, SEC62	2	2.68E-02
hsa04510	Focal adhesion	ITGA9, CAV2, CAV1, CCND2, FYN, PGF, JUN, SOS2, ITGB3, LAMB1, KDR, MYL9	2	3.95E-02

**Table ijerph-17-01373-t001b:** (**b**)

Symbol	Description	Feature
EGFR	The epidermal growth factor receptor	Hub protein
KIT	KIT proto-oncogene receptor tyrosine kinase	Hub protein
IRS1	Insulin receptor substrate 1	Hub protein
KDR	kinase insert domain receptor	Hub protein
BUB1B	BUB1 mitotic checkpoint serine/threonine kinase B	Hub protein
CDH1	Cadherin-1	Hub protein
BUB1	BUB1 mitotic checkpoint serine/threonine kinase	Hub protein
TEK	TEK receptor tyrosine kinase	Hub protein
TPM2	tropomyosin 2	Hub protein
NR4A2	nuclear receptor subfamily 4 group A member 2	Hub protein
FOXC1	Forkhead Box C1	Reporter Transcription Factor
GATA2	GATA Binding Protein 2	Reporter Transcription Factor
YY1	YY1 Transcription Factor	Reporter Transcription Factor
FOXL1	Forkhead Box L1	Reporter Transcription Factor
SPTLC1	Serine Palmitoyltransferase Long Chain Base Subunit 1	Reporter Transcription Factor
E2F1	E2F Transcription Factor 1	Reporter Transcription Factor
CD36	cluster of differentiation 36	Reporter Transcription Factor
SRF	Serum Response Factor	Reporter Transcription Factor
POU2F2	POU class 2 homeobox 2	Reporter Transcription Factor
EGR2	Early growth response 2	Reporter Transcription Factor
hsa-mir-548c-3p	MicroRNA 548c	Reporter microRNA
hsa-mir-335-5p	MicroRNA 335	Reporter microRNA
hsa-mir-26b-5p	MicroRNA 26b	Reporter microRNA
hsa-mir-155-5p	MicroRNA 155	Reporter microRNA
hsa-mir-124-3p	MicroRNA 124	Reporter microRNA
hsa-mir-1-3p	MicroRNA 1	Reporter microRNA
hsa-mir-16-5p	MicroRNA 16	Reporter microRNA
hsa-mir-1-1	MicroRNA 1	Reporter microRNA
hsa-mir-192-5p	MicroRNA 192	Reporter microRNA
hsa-mir-215-5p	MicroRNA 215	Reporter microRNA

**Table ijerph-17-01373-t002a:** (**a**)

Go Term	Pathway	Genes in the Pathway	Fold Enrichment	Adjusted *p*-Value
GO:0006636	unsaturated fatty acid biosynthetic process	ELOVL4, SCD, ELOVL7	45	1.88E-03
GO:0007094	mitotic spindle assembly checkpoint	MAD2L1, BUB1, BUB1B	33	3.39E-03
GO:0045773	positive regulation of axon extension	TNFRSF12A, MAP1B, FN1	24	6.83E-03
GO:0044772	mitotic cell cycle phase transition	CDK1, CKS2	67	2.92E-02
GO:0046602	regulation of mitotic centrosome separation	KIF11, NEK2	67	2.92E-02
GO:0021930	cerebellar granule cell precursor proliferation	ATF5, RORA	45	4.35E-02
GO:0035999	tetrahydrofolate interconversion	TYMS, SHMT2	45	4.35E-02
GO:0035338	long-chain fatty-acyl-CoA biosynthetic process	ELOVL4, ELOVL7	45	4.35E-02
GO:0046777	protein autophosphorylation	EGFR, NEK2, STK26, MELK	5	4.78E-02

**Table ijerph-17-01373-t002b:** (**b**)

Go Term	Pathway	Genes in the Pathway	Fold Enrichment	Adjusted *p*-Value
GO:0045926	negative regulation of growth	ING5, MT1M, MT2A, MT1E, MT1H, MT1X, MT1G, MT1F	18	1.37E-07
GO:0071294	cellular response to zinc ion	MT1M, MT2A, MT1E, MT1H, MT1X, MT1G, MT1F	16	3.17E-06
GO:0006590	thyroid hormone generation	DIO2, CPQ, FOXE1, TPO, DIO1, IYD	21	4.56E-06
GO:0001525	angiogenesis	FGFR2, EMCN, CAV1, NRP1, ACVRL1, EPAS1, PGF, MMP2, KDR, PKNOX1, CTGF, ID1, JUN, MAPK14, TEK, SCR, RHOB, CALCRL	3	1.89E-05
GO:0001938	positive regulation of endothelial cell proliferation	CAV2, NRP1, ACVRL1, PGF, F3, JUN, TEK, ITGB3, CXCL12, KDR	6	2.97E-05
GO:0000122	negative regulation of transcription from RNA polymerase II promoter	FGFR2, GLIS3, TSHZ3, ZMYND11, CAV1, TFCP2L1, ZNF366, PRDM16, CALR, CBX7, CITED2, MINA, TCF4, BHLHE41, NFIL3, NR2F1, TXNIP, EGR1, NR4A2, TLE1, FOSB, SNAI2, FOXP2, CD36, BTG2, ID1, SALL1, FOXE1, LRP8, ID3, PRDM1, SMARCA2, KLF4, NFIA, RERE, PEG3	2	3.16E-05
GO:0010628	positive regulation of gene expression	CAV1, EPHX2, GJA1, TLE1, CDK6, KIT, CALR, CITED2, ANK2, GSN, CTGF, ID1, MAPK14, CYP26B1, NTRK2, RGCC, PRDM1, NFIL3, KLF4	3	4.33E-05
GO:0030198	extracellular matrix organization	B4GALT1, FBN1, OLFML2A, CDH1, ITGB3, KDR, CSGALNACT1, SMOC2, ITGA9, TNFRSF11B, COL9A3, ERO1B, FBLN5, JAM2, LAMB1, CYR61	4	5.55E-05
GO:0043434	response to peptide hormone	BTG2, GNAI1, CTGF, TEK, ANXA1, GJA1, CXCL12, IRS1	8	6.31E-05
GO:0071276	cellular response to cadmium ion	MT1E, MT1H, MT1X, MT1G, MT1F	13	5.36E-04

**Table ijerph-17-01373-t003a:** (**a**) Subnetwork Module 1

Biological Process of Subnetwork Module 1		
Term	*p* Values	Genes
ATP binding	1.00E-13	CDK1, KIF11, CCNB2, NEK2, BUB1, CENPF, BUB1B, NDC80, ANLN, PBK, CEP55, ASPM
Cell division	1.28E-10	CDK1, KIF11, MAD2L1, CCNB2, NEK2, CKS2, BUB1, CENPF, BUB1B, NDC80, UBE2C
Sister chromatidcohesion	6.92E-07	MAD2L1, BUB1, CENPF, BUB1B, NDC80, CENPK
Cell proliferation	1.76E-06	CDK1, TYMS, DLGAP5, CKS2, BUB1, CENPF, BUB1B, MELK
mitotic spindle assembly checkpoint	4.64E-06	MAD2L1, BUB1, CENPF, BUB1B
chromosome segregation	4.68E-06	KIF11, NEK2, CENPF, NDC80, TOP2A
mitotic sister chromatidsegregation	9.32E-06	MAD2L1, NEK2, NUSAP1, NDC80
G2/M transition of mitotic cell cycle	7.46E-05	CDK1, CCNB2, NEK2, MELK, HMMR
negative regulation of ubiquitin-protein ligase activity involved in mitotic cell cycle	2.20E-04	CDK1, MAD2L1, BUB1B, UBE2C
positive regulation of ubiquitin-protein ligase activity involved in regulation of mitotic cell cycle transition	2.69E-04	CDK1, MAD2L1, BUB1B, UBE2C
Cellular Component of Subnetwork Module 1		
Kinetochore	1.39E-07	MAD2L1, NEK2, BUB1, CENPF, BUB1B, NDC80
Condensed chromosome kinetochore	1.99E-07	MAD2L1, NEK2, BUB1, BUB1B, NDC80, CENPK
Nucleoplasm	1.74E-05	CDK1, DTL, EZH2, KIAA0101, CENPF, ANLN, UBE2C, CENPK, TYMS, CCNB2, FANCI, RRM2, BUB1, TOP2A, UBE2T
Nucleus	3.38E-05	CDK1, DTL, NEK2, DLGAP5, EZH2, KIAA0101, NUSAP1, CENPF, NDC80, PBK, CDKN3, CENPK, TYMS, CCNB2, MAD2L1, RRM2, TOP2A, UBE2T, ASPM, MELK
Midbody	4.30E-05	CDK1, NEK2, CENPF, CEP55, ASPM
Cytoplasm	9.43E-05	CDK1, KIF11, DTL, NEK2, DLGAP5, EZH2, KIAA0101, NUSAP1, CENPF, UBE2C, CDKN3, TYMS, FANCI, RRM2, BUB1, BUB1B, TOP2A, UBE2T, ASPM
Centrosome	4.30E-04	CDK1, CCNB2, DTL, NEK2, CENPF, CEP55
Cytosol	5.65E-04	CDK1, KIF11, NEK2, CENPF, NDC80, UBE2C, CENPK, HMMR, TYMS, CCNB2, MAD2L1, RRM2, BUB1, BUB1B
Spindle pole	6.11E-04	KIF11, MAD2L1, NEK2, CENPF
Spindle microtubule	0.0020693	CDK1, KIF11, NUSAP1
Molecular Function of Subnetwork Module 1		
Chromatin binding	3.63E-05	CDK1, EZH2, CKS2, KIAA0101, CENPF, TOP2A, UBE2T
Protein kinaseactivity	2.78E-04	CDK1, NEK2, BUB1, BUB1B, PBK, MELK
protein serine/threoninekinaseactivity	3.44E-04	CDK1, NEK2, BUB1, BUB1B, PBK, MELK
ATP binding	4.78E-04	CDK1, KIF11, NEK2, BUB1, BUB1B, PBK, UBE2C, TOP2A, UBE2T, MELK
Protein binding	9.40E-04	CDK1, DTL, NEK2, DLGAP5, EZH2, KIAA0101, NUSAP1, CENPF, NDC80, PBK, CEP55, UBE2C, CENPK, CDKN3, HMMR, MAD2L1, CCNB2, FANCI, RRM2, BUB1, CKS2, BUB1B, TOP2A, MELK
Ubiquitinconjugating enzyme activity	4.70E-2	UBE2C, UBE2T

**Table ijerph-17-01373-t003b:** (**b**) Module 2 and Module 3

Biological Process of Module-2		
Term	*p* Values	Genes
positive regulation of protein phosphorylation	6.67E-05	EGFR, CCND2, TEK, KDR
peptidyl-tyrosine phosphorylation	1.16E-04	EGFR, BCR, TEK, KDR
protein autophosphorylation	1.64E-04	EGFR, BCR, TEK, KDR
positive regulation of cell proliferation	1.65E-04	EGFR, CCND2, IRS1, KDR, FN1
positive regulation of ERK1 and ERK2 cascade	1.73E-04	EGFR, JUN, TEK, KDR
positive regulation of GTPase activity	3.47E-04	EGFR, BCR, JUN, TEK, IRS1
Angiogenesis	3.52E-04	JUN, TEK, KDR, FN1
positive regulation of fibroblast proliferation	5.48E-04	EGFR, JUN, FN1
positive regulation of endothelial cell proliferation	8.94E-04	JUN, TEK, KDR
transmembrane receptor protein tyrosine kinase signaling pathway	1.72E-03	EGFR, TEK, KDR
Molecular Function of Subnetwork Module-2		
protein tyrosine kinase activity	5.51E-05	EGFR, BCR, TEK, KDR
transmembrane receptor protein tyrosine kinase activity	2.20E-04	EGFR, TEK, KDR
enzyme binding	8.24E-04	EGFR, BCR, JUN, WWOX
protein binding	1.45E-03	EGFR, BCR, CCND2, JUN, TEK, CDH1, TCF4, IRS1, WWOX, KDR, FN1
integrin binding	1.67E-03	EGFR, KDR, FN1
Ras guanyl-nucleotide exchange factor activity	2.00E-03	EGFR, TEK, IRS1
receptor signaling protein tyrosine kinase activity	5.91E-03	EGFR, KDR
identical protein binding	8.26E-03	EGFR, JUN, TCF4, FN1
growth factor binding	1.59E-02	TEK, KDR
chromatin binding	2.13E-02	EGFR, JUN, TCF4
Cellular Component of Subnetwork Module-2		
membrane raft	6.54E-03	EGFR, TEK, KDR
endosome	7.76E-03	EGFR, CDH1, KDR
apical plasma membrane	1.27E-02	EGFR, TEK, FN1
focal adhesion	2.22E-02	EGFR, TEK, CDH1
cell junction	3.00E-02	BCR, CDH1, KDR
microvillus	3.39E-02	TEK, WWOX
Biological Process of Subnetwork Module-3		
protein ubiquitination	3.82E-05	SH3RF1, LMO7, FBXO32, FBXO9
Molecular Function of Subnetwork Module-3		
ubiquitin ligase complex	2.07E-04	RNF144B, LMO7, FBXO9
SCF ubiquitin ligase complex	1.09E-02	FBXO32, FBXO9
Cellular Component of Subnetwork Module-3		
ubiquitin-protein transferase activity	2.89E-05	RNF144B, LMO7, FBXO32, FBXO9
zinc ion binding	2.62E-02	SH3RF1, RNF144B, LMO7

**Table ijerph-17-01373-t004a:** (**a**)

Pathway	Total	Expected	Hits	*p*-Value
Endocytosis	101	0.906	4	1.22E-02
PPAR signaling pathway	64	0.574	3	1.92E-02
Hypertrophic cardiomyopathy(HCM)	25	0.224	2	2.07E-02
Complement and coagulation cascades	67	0.601	3	2.16E-02
Cytokine-cytokine receptor interaction	253	2.27	6	2.40E-02
Adherens junction	70	0.628	3	2.43E-02
Thyroid cancer	28	0.251	2	2.56E-02
Glycine, serine and threonine metabolism	33	0.296	2	3.48E-02
Tyrosine metabolism	33	0.296	2	3.48E-02
Pathways in cancer	310	2.78	6	5.62E-02

**Table ijerph-17-01373-t004b:** (**b**)

Pathway	Total	Expected	Hits	*p*-Value
Pathways in cancer	310	5.37	20	1.66E-07
Chronic myeloid leukemia	73	1.26	8	3.07E-05
MAPK signaling pathway	265	4.59	15	3.67E-05
Maturity onset diabetes of the young	24	0.415	5	4.50E-05
Pertussis	52	0.9	6	2.41E-04
Transcriptional misregulation in cancer	19	0.329	4	2.65E-04
Acute myeloid leukemia	57	0.987	6	4.00E-04
HTLV-I infection	199	3.44	11	5.53E-04
Prostate cancer	87	1.51	7	6.87E-04
Thyroid cancer	28	0.485	4	1.24E-03

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
