# Peer review of "Network-Based Genetic Profiling Reveals Cellular Pathway Differences Between Follicular Thyroid Carcinoma and Follicular Thyroid Adenoma"

_ijerph, 2020, doi:10.3390/ijerph17041373_

Round 1
Reviewer 1 Report
The manuscript "Network-based genetic profiling reveals cellular pathway differences between follicular thyroid carcinoma and follicular thyroid adenoma" by Ali Hossain et al. describes in silico analyses to identify molecular pathways involved in the progression from FTA to FTC.
In general, the manuscript is well written and balanced if considering the value of the data presented and the highlighted need for downstream experimental validation.
However, the authors could provide additional info or comments to make the approach they used more solid and understandable. Specifically:
- In the case of FTC, the authors should provide an indication of the clinical metrics adopted to classify thyroid carcinomas and if they have considered some specific parameters in this regard. This is very important to even understand a sort of "stadiation" of the primary tumor under consideration for the analysis.
- The authors point out that a comparison of FTC to normal thyroid tissues has been already performed. To which extent the molecular pathways identified differ in general in those cases with respect to the description presented in this work? Can the author prioritize such a molecular pathway?
- Are the Hubs identified in Figure 4, corresponding to both protein and mRNA levels? If not, can the author prioritize this info at this level? This would help design specific probes to be tested for downstream validation.
Author Response
Reviewer 1 comment:
The manuscript "Network-based genetic profiling reveals cellular pathway differences between follicular thyroid carcinoma and follicular thyroid adenoma" by Ali Hossain et al. describes in silico analyses to identify molecular pathways involved in the progression from FTA to FTC.
In general, the manuscript is well written and balanced if considering the value of the data presented and the highlighted need for downstream experimental validation.
However, the authors could provide additional info or comments to make the approach they used more solid and understandable. Specifically:
- In the case of FTC, the authors should provide an indication of the clinical metrics adopted to classify thyroid carcinomas and if they have considered some specific parameters in this regard. This is very important to even understand a sort of "stadiation" of the primary tumor under consideration for the analysis.
Response:
We have provided the following description in the Methods section on page 3 to clarify the clinical metrics used by Woytas et al. (2017). The Woytas et al. study describes the following classification approach for tumours using WHO approved criteria (see DeLellis et al, WHO Pathology and Genetics. Tumours of Endocrine Organs; IARC Press: Lyon, France, 2004):
“ the clinical material was divided into primary and secondary sets of tumors, depending on the concordance in histopathological diagnosis. The primary set contains all samples that were independently diagnosed by two thyroid pathology experts …. The secondary set has samples that were diagnosed by only one expert … equivocal samples were diagnosed by two experts ….”
It should be noted that the Woytas et al. study used their classifications to identify tissues only as FTA or FTC. It is thus not clear if they included tissues with uncertain classification. However, because they used the tomour classifications to determine the predictive value of PCR-based markers of FTA/FTC, it is likely that uncertain samples were not used.
Regarding the reviewer’s second point on whether the tumour data could be stratified to provide more information about pathways that may relate to different tumour features, we agree that this would desirable. However, there is little useful information on this available in the dataset, aside from concordance of diagnoses.
Reviewer 1 comment-
The authors point out that a comparison of FTC to normal thyroid tissues has been already performed. To which extent the molecular pathways identified differ in general in those cases with respect to the description presented in this work? Can the author prioritize such a molecular pathway?
Response:
Wang et al (2018) examined on Normal vs. FTC and Normal vs. FTA in the microarray dataset, accession no GSE27155. They found some important pathways including Drug metabolism cytochrome p450, Viral carcinogenesis, Tyrosine metabolism, Cytokine-cytokine receptor interaction, and Cocaine Addition significant pathways for Normal vs. FTC.
In our study, we performed analysis on the FTA vs. FTC tissue data and found several important pathways, as is mentioned in the results section. However, there are no common pathways between our identified pathways that are associated with the FTC to FCA tissues and previously identified pathways that are associated with the FTC to Normal tissues. A tumour may have very different cellular profiles compared to normal tissue (e.g. due infiltrating leucocytes and stromal cells) so here it is better to compare with a benign or non-aggressive tumour such as FTA.
Reviewer 1 comment-
Are the Hubs identified in Figure 4, corresponding to both protein and mRNA levels? If not, can the author prioritize this info at this level? This would help design specific probes to be tested for downstream validation
Response: The Hubs identified in Figure 4 relate to TFs that interact (and so probably regulate expression of) the DEGs. This could in principle be used for probes (e.g., antibodies) that could be used experimentally as part of a validation, however, TF expression is often low, and TF activities often depend on nuclear import, phosphorylation state and binding to accessory proteins. These considerations make their direct use in validation technically problematic, but they are useful to indicate pathways as we have used them here.
Reviewer 2 Report
In submitted manuscript Authors described a new network-based integrative analyses and reported genes, miRNA and pathways which may differ between FTC and FTA. But this work states only network studies. This is a main concern of this paper.
Simply, lack of validation of data by authors of manuscript makes the impact of the published data very limited.
The other concern is number of analyzed samples- total n=52, which is rather low. Author used only a microarray gene expression dataset (Gene Expression Omnibus, n=52) which was generated, carefully analyzed and published in 2017 by Wojtas et al 2017 (Int J Mol Sci. 2017 doi: 10.3390/ijms18061184).
It is important that Wojtas et al 2017 validated their results by quantitative real-time polymerase chain reaction (qRT-PCR) in an independent set of 71 follicular neoplasms from formaldehyde-fixed paraffin embedded tissue material.
Additionally this group also published work in 2013 (BMC Med Genomics 2013 ”Molecular differential diagnosis of follicular thyroid carcinoma and adenoma based on gene expression profiling by using formalin-fixed paraffin-embedded tissues”), which should be mentioned/discussed in submitted manuscript.
Overall in my opinion a novelty of the presented result is limited.
Author Response
Reviewer 2 comments:
In submitted manuscript Authors described a new network-based integrative analyses and reported genes, miRNA and pathways which may differ between FTC and FTA. But this work states only network studies. This is a main concern of this paper.
Simply, lack of validation of data by authors of manuscript makes the impact of the published data very limited.
Response:
It is important to note that the focus of this work is to identify pathways that are important in the pathogenesis and behavior of these tumours. Finding such pathways can give new insights into these tumours and, in general, these pathways cannot be identified by other means. This can provide important information for the design of functional studies (which can lead to drug design) particularly if particular pathways involved are implicated in other disease processes. Most proteins in a signaling pathway (for example) are not regulated at the mRNA level but by phosphorylation and protein-protein interactions. Thus, it is simply not possible to validate such pathway data in any simple way, as this would require a great deal of work with a functional model (eg an animal model of human FTC) where blocking a particular pathway affects tumour behavior. Simply measuring the expression of a particular gene in FTC tumours (for example) would not constitute validation here, although it might be useful to further confirm the array data.
Reviewer 2 comments:
The other concern is number of analyzed samples- total n=52, which is rather low. Author used only a microarray gene expression dataset (Gene Expression Omnibus, n=52) which was generated, carefully analyzed and published in 2017 by Wojtas et al 2017 (Int J Mol Sci. 2017 doi: 10.3390/ijms18061184).
It is important that Wojtas et al 2017 validated their results by quantitative real-time polymerase chain reaction (qRT-PCR) in an independent set of 71 follicular neoplasms from formaldehyde-fixed paraffin embedded tissue material.Additionally this group also published work in 2013 (BMC Med Genomics 2013 ”Molecular differential diagnosis of follicular thyroid carcinoma and adenoma based on gene expression profiling by using formalin-fixed paraffin-embedded tissues”), which should be mentioned/discussed in submitted manuscript.
Response:
In Woytas et al. (2017), the RT-PCR was used to study the relationship of the putative markers to the clinical results. Woytas et al. (2017) differentiate between two histologically defined tumour entities. This is very different from the focus of our work, which is to find clues that point to pathways (and pathway hubs) that differ between the two types of tumour, taking advantage of the massively parallel nature of the array data. The accuracy of data on individual DEGs was less important than finding evidence for the pathways. The latter data were then filtered and examined for other evidence of plausible pathogenic mechanisms as described in the methods section.
The Pfeifer et al (BMC Med Genomics 2013) study used publically available array data (from Borup et al, 2010, EndocrRel Cancer 17:691) to identify markers to differentiate FTC and FTA and used PCR approaches as a machine learning (SVM) classification approach to identify markers. We have mentioned in the discussion section (p 12) some points about this work.
Reviewer 2 comments:
Overall, in my opinion, a novelty of the presented result is limited.
Response:
Our approach is not really a conventional one, as it attempts to take advantage of the complexity of the disease (i.e., the large number of proteins with altered expression) that is usually an obstacle to identifying individual pathogenic genes/ and proteins. As noted above, it is also quite different from the search for diagnostic markers that was the focus of Woytaset al. (2017) and Borup (2010) papers. Our work uses an analysis pipeline that we developed, and identifies potential factors that have roles in the development and behaviour of the genes. Validating our results biologically would involve blockade of the hub proteins, TFs or miRNAs and observing the consequences. In practice, this is advanced using evidence from studies that identify these factors as pathogenic in other diseases to clarify the functions of these factors and pathways. The novelty of our work lies in the way we have used the data to find factors of interest by methods quite different from Woytas and co-workers, and for a different purpose, i.e. to find critical pathways in FTC. Noone to our knowledge has done this, and it is quite novel.
Round 2
Reviewer 2 Report
Dear Authors. Thank you for responses/comments.
I have decided to recommend the manuscript for publication in the present form.